# Making the Unsustainable Sustainable: How Swedish Secondary School Teachers Deal with Sustainable Development in Their Teaching

**Per Gyberg** [1,*] **, Jonas Anshelm** [2] **and Jonas Hallström** [3]

1   Department of Thematic Studies—Environmental Change, Linköping University,
    S-581 83 Linköping, Sweden
2   Department of Thematic Studies—Technology and Social Change, Linköping University, S-581 83 Linköping,
    Sweden; jonas.anshelm@liu.se
3   Department of Behavioural Sciences and Learning—Division of Learning, Aesthetics, Natural Science,
    Linköping University, S-581 83 Linköping, Sweden; jonas.hallstrom@liu.se
*   Correspondence: per.gyberg@liu.se; Tel.: +46-(0)13-28-5616

**Abstract:** The aim of this article is to investigate how Swedish teachers manage the uncertainty and complexity associated with sustainable development (SD) as a field of knowledge, in relation to the requirements in the school curriculum. Underlying the whole concept of sustainable development is the vision that there is a possible solution to the ecological, economic and social problems created by humans. However, it is not so clear what this solution actually means in practice. The article builds on an analysis of transcribed individual and group interviews with 40 teachers at Swedish lower and upper secondary schools, related to the topic of sustainable development as a field of knowledge. A thematic analysis was carried out by identifying four broad themes, including dominating discourses. The results indicate that there is a lack of vision among the teachers for a future sustainable society, while at the same time, it seems to be taboo to talk about what an unsustainable society might mean in the long run. Presentations of the problems and knowledge of what causes them must always be combined with instructions on how problems can be solved and how pupils can influence their own future and help create sustainable development. The starting point for such a solution-oriented approach to SD is based on an assumption that individual behaviour is essential to achieving sustainable development and thus that individual responsibility is crucial. This focus leads to individual consumer choices, behaviours and lifestyles at the heart of teaching, while progressive, alternative visions and critical perspectives are downplayed.

**Keywords:** sustainable development; discourse; future citizens; ecological modernisation

## 1. Introduction

The concept of sustainable development refers, on the one hand, to the vision of a possible solution to the ecological, economic and social problems that humanity has created. On the other hand, the concept also implies the opposite: the risk of unsustainable development, if proper measures are not undertaken. Unsustainable development entails a dystopic future where the preconditions for human, as well as non-human, existence are threatened. In spite of—or, perhaps, because of—the plasticity and ambiguity that the concept has turned out to involve, it offers no clear vision of what a sustainable society might look like in the future [1]. However, it is clear that many of the identified problems in relation to sustainable development remain to be resolved, such as the all-eclipsing issue of global warming. Many of these problems also involve seriously questioning how, in particular, the rich part of the world has arranged its societies and lifestyles [2]. This means that, despite its plasticity,

the concept—and the problems it implies—also has an inherent energy that includes a threat to the prevailing social order [3]. The concept includes a demand for change.

In a global perspective, education is singled out as one of the key means by which development can become more sustainable [4–7] The UN plays a central role in which education is seen as one of the main tools for achieving sustainable development. Resolution 57/254, Decade of Education for Sustainable Development, adopted in 2002, gives the term a more central role and the objective is to "[ . . . ] integrate the principles, values, and practices of sustainable development into all aspects of education and learning" [8]. A great deal has happened since then, and as far as Swedish education is concerned, the concept is a feature of primary, lower secondary and upper secondary curricula and the course syllabuses for many subjects. The concept is part of schools' fundamental values and tasks: "Teaching should illuminate how the functions of society and our ways of living and working can best be adapted to create sustainable development," according to the definition of "environmental perspective" [9]. The concept is also explicitly expressed in the objectives, core content and/or various knowledge requirements for the majority of subjects.

It is not a coincidence that education occupies such a prominent position in addressing the complex problems related to sustainable development [6,10–13]. Rather, it reflects the strength of the confidence in education's ability to get people to think and act differently through knowledge, and the fact that it involves a long-term change process. At the same time, this means that education must address both the plasticity of the sustainable development concept and its inherent criticism of the prevailing order [14]. However, the compulsory school and upper secondary school curricula do not give any clear directive for this purpose, despite the concept's central position in the steering documents [9,15].

Education that touches on problems related to the relationship between humans and the environment has changed from dealing with a biologically governed definition of environmental problems to an approach that emphasises ethical and societal aspects [6,7,16–20]. Accordingly, it has gone from environmental teaching to education for sustainable development, which is regarded as broader, and thus stretches the boundaries of the field of knowledge far beyond those of any individual subject. In addition, the field of knowledge does not only have an inherent potential for change—its very existence in course syllabuses and teaching materials also implies a clear demand and an expectation for change: "Sustainability [ . . . ] refers to the process or strategy of moving toward a sustainable future" [21–23].

At the same time, research studies show that teachers see the environmental field as increasingly complex, and that what counts as being environmentally friendly is more and more being called into question [10,19,24–27]. The complexity of this and similar fields of knowledge means that there is a danger of losing sight of the bigger picture. One example of this is the way in which teaching and teaching materials relating to the environment and sustainable development often have a solution-oriented character [28]. Recycling, turning out lights or knowledge about different environmental and fair-trade labelling systems are examples of this, as the causes of the problems may be trivialised when they are seen to be solvable with simple behaviours. The bigger picture is not only thought to be difficult to incorporate into teaching, but is also often conspicuous by its absence [29,30]. This is in spite of the fact that curricula and course syllabuses in many countries indicate the intention to make a transition from fact-based teaching to teaching that incorporates authentic problems, cross-curricular issues and contexts [31–33]. Many teachers feel that they simply do not have the time to work in a more authentic and interdisciplinary manner [25]. Borg et al. [34] found that just 25% of Swedish upper secondary school teachers take an interdisciplinary approach when working with sustainable development, because the overall pictures, complexities, different knowledge forms and norms it includes must be translated into knowledge requirements and ultimately grades, or—and this will be the focus of this study—into well socialised, responsible and knowledgeable future citizens. The latter has been addressed by Ideland and Malmberg who, in their study of compulsory school textbooks, show the expectations placed on a 'sustainable' person: "According to the books, this person

should be civilised, organised, clean, technologically knowledgeable and helpful; in short—a modern person" [28].

The aim of this article is to investigate how Swedish teachers manage the uncertainty and complexity associated with sustainable development as a field of knowledge, in relation to the requirements in the school curriculum. We will do this by analysing how Swedish secondary school teachers portray future citizens and a future sustainable society in interviews. Our investigation is guided by the following research question: According to teachers, what qualities are future citizens and a future sustainable society expected to have? Investigating teachers' perspectives on sustainable development is crucial because of the importance of teachers' attitudes and knowledge for developing pupils' attitudes [35]. More specifically, this study could contribute to the understanding of how ideas about future citizens and a future society are entangled with, and potentially limit, teaching about sustainable development.

## 2. Theoretical Framework and Methodology

In this study, we are interested in how the view of the individual is formed in relation to sustainable development and the individual's role in a potential future sustainable society. Our view of how teachers' descriptions and interpretations should be understood and interpreted is inspired by Foucault and a discourse theory tradition [26,36]. We believe that what teachers say about the citizen of the future is an accentuation of possible subject positions within existing discourses [36,37]. With this approach, the discourse or discourses on "the environmentally aware citizen" offer a number of possible ways of being, while at the same time excluding others. In the spirit of Foucault, a discourse is understood here as a productive system of meanings. The discourse is productive because it creates a certain meaning at the same time as it also excludes other meanings [36,37].

The data of this study were collected through semi-structured interviews, which were recorded and transcribed. The interviews were carried out with 40 Swedish teachers in secondary education, thirteen of whom were lower secondary school teachers and 27 upper secondary school teachers. A total of seventeen different schools were involved: eleven upper secondary schools and six lower secondary schools. Group interviews involving three or more people were conducted on eight occasions, and the remainder were interviewed individually. The same interview guide was used for the group interviews as for the individual interviews. The reasons for choosing group interviews on certain occasions were primarily practical. Generally speaking, it was hard to get teachers to take part, and a lack of time was usually given as the reason. However, some schools suggested that the interviews could be carried out in groups and at a time when the teachers would be involved in some form of joint development or planning work anyway. This option was therefore chosen in some schools when they declined individual interviews (see Table 1).

**Table 1.** Teachers, schools, and type of data collection in the study.

| Teachers/Schools | Lower Secondary School | Upper Secondary School | TOTAL |
|---|---|---|---|
| Teachers | 13 | 27 | 40 |
| Schools | 6 | 11 | 17 |
| **Data collection** | - | - | - |
| Group interviews | 3 | 5 | 8 |
| Individual interviews | 3 | 13 | 16 |

Semi-structured interviews were chosen because this is an explorative approach and the desire was not to define in advance what sustainable development is and how it is included in teaching [38,39]. In that respect, it was also important to allow the teachers the greatest possible opportunity to express their own experiences and thoughts. Although the interview guide contained certain themes and questions under each theme (generally relating to sustainable development, sustainable development

in education and the citizens of the future), no crucial importance was attached to standardising each individual question. Instead, the questions were largely adapted to the individual interviewees [38,39]. The group interviews were therefore also like focus group interviews, in that the teachers themselves had the opportunity to steer the issues discussed, and to a certain extent, these interviews were thus even less guided by the researcher than the individual interviews [38–40].

We originally focused on teachers in larger subject areas for which sustainable development is most strongly emphasised in the curricula for Swedish lower secondary school and upper secondary school, i.e., STEM subjects (in Swedish schools, science, technology and mathematics), social sciences and home economics. However, other teachers—in languages, sport and health, and art—also became involved in our contact with the various schools, and were thus included in the study. When contacting some schools, we were referred to certain teachers who were regarded as being particularly "involved" in sustainable development, and, in other cases, the teachers themselves brought along a colleague to one of the group interviews. This made the data both broader and narrower than planned. The data became broader in the sense that roughly an equal distribution of men and women in a wide spectrum of subjects relating to sustainable development in both lower and upper secondary education took part in the study. The data became narrower in the sense that it may be that it was especially those teachers who were interested in or believed they had experience of teaching sustainable development who agreed to take part in the study. Although this sort of skewness must be taken into consideration, several teachers also expressed the opposite, or chose to participate in the study because they were asked to by colleagues or the head teacher. Finally, because of the general focus on sustainable development (SD), future citizens and a future sustainable society, we chose not to differentiate between gender, subject area, or level of education among the 40 teachers when presenting the data.

A systematic read-through and analysis were carried out of the entire data set of transcribed interview material in several phases, and four broad themes were identified [41]. The identification of themes was conducted first by being attentive to patterns in how the teachers spoke about how they handle the phenomenon sustainable development in their teaching [42]. For each generated theme, it was a case of identifying key meanings and investigating how these were organised and built up. In a final phase, we connected the teachers' utterances under the four themes to dominating discourses, as presented under the theoretical framework above. Under each theme, we present particularly illustrative quotations as examples that included such discourses, although, due to the many interviews, there was only space enough for a few respondents to be represented.

The study data features potential limitations in that there is a degree of asymmetry in terms of (1) the different interview formats, and (2) which teachers are represented in the study, as the majority were upper secondary teachers (see Table 1). Since the interviews were explorative with a great deal of freedom for the respondents, the choice of interview format should not affect the results, and we have tried not to generalise either lower or upper secondary school teachers' perceptions of sustainable development. Both lower and upper secondary schools feature a strong subject tradition, and it is this very link between sustainable development, teachers and several broad school subject areas that is the focus of our study, not specifically the level of education. Given these potential limitations, the empirical material gives a good picture of how teachers deal with sustainable development in relation to the aim of the study.

As promised when the interviews were conducted, both the teachers and the schools were anonymised. In order to meet the requirements of research ethics [43], therefore, the participating teachers were informed on each individual instance of data collection about the aims of the study and that the consent to participate that they had given could be withdrawn at their own request. It was also explained to the teachers how the data material for the study would be handled according to Swedish and EU regulations (e.g., The General Data Protection Regulation) and that the data would be used only for research purposes. The anonymity of the participating teachers is also ensured by the neutral way we present the interview transcripts.

## 3. Results

### 3.1. Aware and Involved Citizens

Virtually all the informants emphasised that the aim of education for sustainable development is to create involvement, awareness and ultimately a change of attitude. One question that follows on from this is what the pupils should be involved in and aware of. In particular, the human role in ecosystems was highlighted as being central; the understanding that humankind is part of nature and is dependent on interaction with it was emphasised, as were the potential consequences if human actions upset the balance of these ecosystems. One teacher expressed it this way:

> I want to arouse some kind of feeling that this is actually important, some kind of involvement. [ ... ] As a teacher, I want my pupils to have a kind of fascination, and as a biologist there's a wealth of biological diversity, and about the ecosystem and how everything fits together, and to discover diversity. [ ... ] We can't build as something outside how nature works—society is also nature; this insight in some way, and how we've got this feeling that this is important and that we're part of this, then I feel it doesn't matter much what we actually work with in purely practical terms within the environment and sustainable development, but it can be very much that we can do. It's usually the environmental issues that come into play then, the climate issue, the Baltic Sea, environmental toxins and ... or things that happen [locally] or whatever, and it's those kinds of things that the group and I agree on that I think are important and that can make us feel involved. After all, the aim is to get the pupils involved, and they won't get involved in the environment unless they have this insight and feeling that this is important and worth preserving [ ... ].

In this case, it is the emphasis on the impossibility of making a distinction between society and nature to generate involvement that is important, but it is the preservation discourse in particular that is central. According to the teacher, teaching should work to evoke a feeling among the pupils that humankind is dependent on nature, and that society is a part of nature. The teacher described this insight as a feeling and believed that this feeling is absolutely crucial to the pupils' sense of involvement; teaching should involve clarifying our dependence on nature.

However, the dependence on nature was not usually raised as an argument in the interviews. Instead, it was the distinction between society and nature that was taken as a starting point, and the fact that nature's limits have been reached was emphasised. This awareness then involves the pupils' understanding this and involvement being created through a desire not to exceed these limits. This other discourse, therefore, does not appeal to a "feeling", but has significantly more rational starting points, as the limits are measurable and controllable. Involvement and awareness involve knowledge of the effects our own behaviour has on whether or not these limits are exceeded.

However, regardless of discourse, the interviewed teachers almost always agreed that this awareness should not be too heavy a burden for the pupils to bear. There is a risk that depictions of misery, pessimism and a pedagogy of disaster will result in dejection. Dismal descriptions of the work involved with sustainable development are not seen to lead anywhere, instead resulting in apathy and resignation. One teacher put it the following way:

> [ ... ] wherever you turn, there's this propaganda of fear all the time. The subject [SD] risks becoming almost too heavy. You have to keep this in mind all the time, so that it doesn't become a case of "It doesn't matter, there's nothing we can do anyway". Instead, you have to keep, yes be positive and see that actually this is an opportunity to make a difference and yes, create new, green jobs that we tend to talk about, too.

The quotations above show the balancing act that teachers feel they have to perform in their teaching. Environmental problems must be highlighted in the classroom, while at the same time, the teachers want to avoid bringing up excessively troublesome future scenarios, although most

teachers probably would not go as far as to call this "propaganda of fear" (Swedish, skräckpropaganda). Of course, the effects of environmental problems are the entire reason for striving to achieve awareness and involvement, but, at the same time, suggesting in some way that it is not possible to do anything about these problems would make the intention of the teaching meaningless.

### 3.2. Solution-Oriented Teaching

According to a solution-oriented approach, presentations of problems and knowledge about what causes them must always be combined with directions for how the problems can be resolved and how the pupils themselves can influence their future or contribute to bringing about sustainable development. Many of the teachers pointed out that the effects of environmental problems should not be something abstract in the future, but must instead be related to pupils' everyday lives, to concrete things. The teacher below said that s/he collected the pupils' litter that s/he had managed to get hold of in various ways over the course of two weeks, and spread it out in the classroom:

> I fully believe, I'm entirely convinced that we have to concretise things, we have to use examples of the environment that the pupils can relate to, before they can relate to other more abstract concepts, and only then can we deal with theoretical aspects. [ . . . ] And this is concrete, but we can't concretise the future. But we can concretise the present. We can abstract it to the future.

This case related to how pupils should experience the effects of their own waste in order for the situation to become concrete, but it also related to the pupils' seeing that they themselves create problems and they themselves can do something about it. At another school, a climate week was planned to raise pupils' awareness of what they themselves do that affects the climate:

> Yes, we'll have [a meteorologist] who comes here and gives a talk, and we'll carry out an environmental quiz walk, we'll involve the school canteen staff and they'll arrange a vegetarian day. There will be an electricity-saving day, and this because Earth Hour will be on Saturday the 29th, so we'll have to hold our own Earth Hour day, so some people will be informing about that, and we'll have a swap-meet day, too.

The solutions are concrete in the sense that they are organised based on specific areas, such as littering, electricity use, recycling and food. Within each action area, there are specific actions where a clearly demarcated, often measurable environmental benefit can be achieved.

Concretisation may primarily mean that a change towards a more sustainable society is presented as being achievable. In the words of one teacher:

> I think we should start with the little things in some way. After all, the pupils aren't too stupid to see that if the world's leaders can't agree to do things, the situation feels a little hopeless. [ . . . ] But if we start to make them aware as I said before, they might not buy apples that have been transported from Chile or wherever, then they still feel that they're doing something. And I think we have to start where we can, we can't bring about a change immediately [ . . . ] and I don't think we can get through to them otherwise. So you put yourself at the centre: What actually affects me? I might be completely wrong, but I think it's easier to start with yourself rather than deciding that you have to save the entire world. In other words, start with what affects me and then extrapolate from there. It's actually also better for the environment.

Change must not be something that is hopeless. Instead, pupils should be inspired by seeing for themselves that they can make a difference. One very strong, recurring idea is the distinction between what the above teacher calls "the little things" and "the big things". The idea of the little things explains why change takes time, that the solutions presented are only to be seen in the very long term, and that every individual's actions contribute towards something bigger.

### 3.3. Individualised Driving Force

The starting point for the solution-oriented approach is an assumption that individual action is crucial for achieving sustainable development, and thus also that individual responsibility is of decisive importance. Through teaching, pupils should be nurtured to become the responsible individuals that sustainable development requires. This focus leads to individual consumption choices, behaviours and lifestyles being placed at the centre of the teaching. Here, the choices of pupils, their families and Swedish citizens can be problematised, and changed behaviours and consumption patterns can be highlighted as a positive solution, in order to achieve a sustainable society. In view of the teachers' intention to encourage action and to highlight pupils' opportunities to contribute towards problem-solving, it seems almost inevitable that the teaching should be targeted at what lies within the framework of the pupils' immediate reach and to communicate that everyday personal choices are of crucial importance. According to one teacher:

> [ . . . ] the idea is that they should be able to evaluate their choices and make considered choices, and in my classes they should be able to justify their choices. Of course, they don't have to justify all their choices, but they should still keep that in mind and understand that making a particular choice is smarter.

Teaching the citizens of the future will thus largely involve nurturing the consumers of the future, while structural, socioeconomic conditions are taken for granted and remain unproblematised. One of the teachers wanted to emphasize:

> That [the pupils] are reflective as consumers, and so on. They already consume a great deal, and will certainly consume even more once they have jobs and earn more money. So it's the aspect of them thinking a bit about where clothes come from and, yes, where food comes from. If we can instil that in them.

The teaching that the teachers described is thus based on a prevailing discourse of continuous economic growth and a notion that established political and economic power structures cannot be influenced or changed; these conditions are beyond the individual sphere of influence. Questioning them would risk undermining the teachers' declared intentions not to make the teaching too "heavy". Instead, according to the teachers, the teaching is firmly based on the assurance that problems relating to sustainable development can be solved with the help of market mechanisms, and that there is actually no alternative to this. It is thus entirely in line with the dominant neoliberal discourse about how social change should be achieved [44]. Questioning this order within the framework of teaching about sustainable development is not possible, as it would risk coming into conflict with the underlying ambition to communicate that pupils have the ability to resolve these problems through their own individual choices. Thus, there is a double message: that pupils' individual choices will solve environmental problems, although the teachers actually believe that this must be achieved structurally through market mechanisms.

### 3.4. An Apolitical Classroom

Some teachers saw problems with socioeconomic and political power structures not being highlighted in the teaching, however, and they thus questioned the dominating discourse. For example, they stated that they would like to problematise the primacy of growth and introduce questions about "de-growth". The reason they cited for actually not doing so in the end is that it would make their teaching too "political", as is evident in the following extract from a group interview with a dialogue between two teachers:

> To establish the historical perspective, a little bit so that it doesn't become too political. We talk nuclear power, to some extent, that is.

> And we think the same thing, that it's included in so many course syllabuses, the concept of growth.

Yes.

I'd like to introduce more about de-growth and other ways of thinking.

Yes.

But time is against us, in some respects.

After all, they're only 18 years old, so they're not so . . . you have to enhance their existing knowledge somehow, and that's not easy with these fairly difficult issues. But then . . . But not perhaps so that we avoid it [ . . . ]

They thus make an active choice to downplay such perspectives, because they do not see them as entirely accepted and suspect that they could be perceived by colleagues, school management, pupils and parents as being provocative. It is also intimated in the extract above that the pupils would not be mature enough to deal with more extensive criticism of the prevailing social order. The logic of the prevailing neo-liberal discourse and the proposed solutions to sustainability issues that this implies are thus understood as being unproblematic, apolitical and neutral. Any challenges to this logic are understood as being too political and thus unsuitable for inclusion in teaching. The dominance of this understanding of what belongs within the field of knowledge of sustainable development is seen not least in those who express a degree of criticism towards it; they do not believe that it as possible or appropriate to introduce challenging perspectives into teaching. Only one of the teachers stated that they had explicitly questioned the individualisation of environmental responsibility in the teaching, and noted that ecological breakdowns were mainly due to socioeconomic structural conditions far beyond individual control:

What do you think the most important thing is—you've already touched upon it—that the pupils take on board if you see that you're teaching the citizens of the future? What? (Interviewer, I)

Yes, that they should learn to think.

Learn to think. (I)

Hmm, consider the consequences of things. What I actually also say to them: 'Don't have a bad conscience!' I think it's important. If you like cars, yes, but then you accelerate like crazy when driving. It doesn't matter, because that's not the problem from an overall perspective. It's like a drop in the ocean. It's the big systems we need to look at. Whether or not you like cars, but accelerate a bit quickly, yes that's fine. So, we leave the minutiae and concentrate on the bigger picture.

One consequence of so few teachers expressing similar perspectives like the one above, is that teaching conveys an almost unequivocally negative image of what sustainable development is or can be. It is consistently described as being associated with self-denial and self-sacrifice for the sake of other people, future generations, other cultures or other species. It is almost always based on helping someone else, and is always achieved through individuals taking responsibility and abstaining. Teachers stated that they try to change pupils' attitudes and engage them in this form of taking responsibility.

## 4. Discussion

The results of this study show, through the four themes, that most teachers want to avoid using a "propaganda of fear" in their teaching. The negative effects of environmental problems are, of course, a reason to strive for awareness and involvement in connection with sustainable development, but the teachers also suggest that, if they signal that we cannot do anything about these problems in our everyday lives, the intention of the teaching will become meaningless. One recurring idea is thus the distinction between what one teacher calls "the little things" and "the big things". The idea of the little things explains why change takes time, and that the solutions presented are only seen in the very long term, and that every individual's actions contribute towards something bigger. Paradoxically,

the teaching is—as deduced from the teachers' descriptions—also firmly based on a discourse of sustainable development that promises solutions with the help of market mechanisms, and that there is actually no alternative to this. Questioning this order within the framework of teaching about sustainable development is not possible, as it would risk coming into conflict with the underlying ambition to communicate that pupils have the ability to resolve these problems through their own individual choices.

One consequence of so few teachers expressing critical perspectives is that teaching conveys an almost unequivocally negative image of what sustainable development is or can be. SD is consistently described as being associated with self-denial and self-sacrifice for the sake of other people, future generations, other cultures or other species. This is also reflected in the dominance of a "limits of nature" discourse, at the expense of a preservation discourse that emphasises holism and the dependence of humans on nature in a more direct sense. SD is thus almost always based on helping someone else, and is achieved through individuals taking responsibility and abstaining. Teachers state that they try to change pupils' attitudes and engage them in this form of taking responsibility. Despite the teachers saying that they want to avoid the heavy, depressing and negative issues, their descriptions of the teaching field of sustainable development are thus almost entirely lacking in positive images of the sustainable society and a future alternative to the prevailing system.

One effect of the predominant discourse is that it excludes the joint creation of a healthier, fairer, more successful, more democratic and more ecologically balanced sustainable society. The sustainable society is never described as a possibility, or as a utopia in a positive sense, but as a forced necessity that requires changes to individual behaviours. The ecological question, for example, is never understood as a gift to forces for social change that could have a liberating effect [45]. The possibility that sustainable development could represent the path towards a socially, culturally and existentially—if not materially—richer life does not feature within the discourse. That it could include a progressive force for social change is excluded by definition. "Transition towns", "de-scaling", a cooperative, sharing economy, eco-towns and other social, ecological and economic experiments are therefore also excluded. In this way, teaching about sustainable development will ultimately be seen in strongly social conservative terms. It contributes, in the words of Ingolfur Blühdorn, to "sustaining the unsustainable" [46]. It advocates, via the manner of the discourse on ecological modernisation [47], a route for marginal corrections within the framework of the dominant economically liberal discourse, to contribute towards avoiding a collapse of the global ecosystems. It thereby also dismisses the opportunity for more sweeping changes to social and living conditions. System-immanent solutions are, in this context, understood as unproblematic, constructive and above all a-political, while all forms of system-transcendent solutions are weeded out as being too political, as if there was a disinterested position. This also means that the extensive criticism of contemporary living conditions is excluded [36,37,44,48].

Thus, according to the informants' implicit logic, teaching about sustainable development is not only deeply politicised. It also entirely excludes self-reflection on this point, and works according to the idea that there is a prevailing consensus that established socioeconomic and political conditions should be the obvious—or even only—starting point for teaching within this field in Swedish schools [49]. The reason for this is probably that teaching on sustainable development is complex, as it involves not only nature, but also the whole "technosystem" of societal markets and administrations and the technologies that they involve [44]. It is easier to assume that pupils—and teachers themselves—can act in the here and now, rather than challenging the prevailing neoliberal social, economic and political order [10,26,27].

## 5. Conclusions

This study shows that the participating teachers lack a positive vision for a future sustainable society, at the same time as it seems to be taboo to talk about what an unsustainable society might actually mean in the long run. Furthermore, presentations of environmental problems—and knowledge

about their causes—always need to be combined with instructions for solutions, which means that the teachers thereby impose on their pupils self-sacrifice for the sake of other people, future generations, other cultures or other species. Thus, more progressive knowledge and actions that may be highly important for contributing to a future sustainable society are not raised; for example, de-scaling, eco-towns and other alternative societal visions. System-immanent solutions are in this context understood as unproblematic, constructive and above all apolitical, while all forms of system-transcendent solutions are weeded out as being too political. Ironically, teaching about sustainable development thereby becomes politicised. There is therefore also a lack of critical perspectives on what knowledge about SD is in Swedish schools, and perspectives that do not support a dominant liberal discourse tend to be excluded.

**Author Contributions:** Conceptualization, P.G.; methodology, P.G., J.H.; formal analysis, P.G., J.A., J.H.; investigation, P.G., J.A., J.H.; writing—original draft preparation, P.G., J.A.; writing—review and editing, P.G., J.H. All authors have read and agreed to the published version of the manuscript.

**Funding:** This research was funded by The Swedish Energy Agency, grant number 32619-1.

**Acknowledgments:** Many thanks to Karin Skill, Fredrik Envall and Hanna Sjögren for critical reading and input.

**Conflicts of Interest:** The authors declare no conflict of interest. The funders had no role in the design of the study; in the collection, analyses, or interpretation of data; in the writing of the manuscript, or in the decision to publish the results.

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
