# Peer review of "Making the Unsustainable Sustainable: How Swedish Secondary School Teachers Deal with Sustainable Development in Their Teaching"

_sustainability, doi:10.3390/su12198271_

Round 1
Reviewer 1 Report
A very interesting study, mostly well written, and the findings are important for teachers and sustainability education in schools.
There are though some improvements that needs to be done:
- Let the readers know from the beginning where from the teachers are (Sweden) and in what school level (secondary school teachers) – consider the title or the first sentence for this.
- In Line 35 it states “the concept also implies the opposite: the risk of unsustainable development.“ Describe what you mean by this.
- The literature needs to be updated, many are a bit old, the newest from 2017.
- Discuss and/or argue for why this study is important.
- Describe better the methods used (were the interviews recorded and transcribed??) especially the analysis (was it according to the six steps of Brown and Clarke, 2006 or how was it done??).
- In lines 194-195 it says: „Virtually all the informants emphasised that the aim of education for sustainable development is to create involvement, awareness and ultimately a change of attitude (cf. Ideland & Malmberg, 2014, p. 383).“ Why do you quote another reference when describing your findings?
- Conclusions are missing. How can these findings contribute to a better sustainability education in school and what other implication can it have?
Author Response
We agree with reviewer 1 and think that it is valuable and important comments that will improve the paper.
- Let the readers know from the beginning where from the teachers are (Sweden) and in what school level (secondary school teachers) – consider the title or the first sentence for this.
Title: Making the unsustainable sustainable: How Swedish secondary school teachers deal with sustainable development in their teaching
First sentence in abstract (line 13): “The aim of this article is to investigate how Swedish teachers manage the uncertainty…”
- In Line 35 it states “the concept also implies the opposite: the risk of unsustainable development.“ Describe what you mean by this.
Line 36-38: “the risk of unsustainable development, if not proper measures are taken. Unsustainable development entails a dystopic future where the preconditions for human as well as non-human existence are threatened.”
- The literature needs to be updated, many are a bit old, the newest from 2017.
- Meisert, A. & Böttcher, F. (2019). Towards a discourse-based understanding of sustainability education and decision making. Sustainability, 11(21), 5902.
- Sund, P. & Gericke, N. (2020). Teaching contributions from secondary school subject areas to education for sustainable development – a comparative study of science, social science and language teachers. Environmental Education Research, 26(6), 772-794.
- Misiaszek, G. W. (2019). Ecopedagogy: teaching critical literacies of ‘development’, ‘sustainability’, and ‘sustainable development’. Teaching in Higher Education, 25(5), 615-632.
- Discuss and/or argue for why this study is important.
Line 109-113: “Investigating teachers’ perspectives on sustainable development is crucial because of the importance of teachers’ attitudes and knowledge for developing pupils’ attitudes (e.g. Nordlöf, Hallström & Höst, 2019). More specifically, this study could contribute to the understanding of how ideas about future citizens and a future society are entangled with, and potentially limit, teaching about sustainable development.”
- Describe better the methods used (were the interviews recorded and transcribed??) especially the analysis (was it according to the six steps of Brown and Clarke, 2006 or how was it done??).
Line 125-126: “The data of this study was collected through semi-structured interviews, which were recorded and transcribed.”
Line 169-178: “A systematic read-through and analysis were carried out of the entire data set of transcribed interview material in several phases, and four broad themes were identified (cf. Braun & Clarke, 2006). The identification of themes was conducted first by being attentive to patterns in how the teachers spoke about how they handle the phenomenon sustainable development in their teaching (Kvale, 1999). For each generated theme, it was a case of identifying key meanings and investigating how these were organised and built up. In a final phase, we connected the teachers’ utterances under the four themes to dominating discourses, as presented under theoretical framework above. Under each theme, we present particularly illustrative quotations as examples that included such discourses, although due to the many interviews there was only space enough for a few respondents to be represented.”
- In lines 194-195 it says: „Virtually all the informants emphasised that the aim of education for sustainable development is to create involvement, awareness and ultimately a change of attitude (cf. Ideland & Malmberg, 2014, p. 383).“ Why do you quote another reference when describing your findings?
Line 201: Quote removed.
- Conclusions are missing. How can these findings contribute to a better sustainability education in school and what other implication can it have?
“Conclusions
This study shows that the participating teachers lack a positive vision for a future sustainable society, at the same time as it seems to be taboo to talk about what an unsustainable society might actually mean in the long run. Furthermore, presentations of environmental problems - and knowledge about their causes - always need to be combined with instructions for solutions, which means that the teachers thereby impose on their pupils self-sacrifice for the sake of other people, future generations, other cultures or other species. Thus, more progressive knowledge and actions that may be highly important for contributing to a future sustainable society are not raised, for example, de-scaling, eco-towns and other alternative societal visions. System-immanent solutions are in this context understood as unproblematic, constructive and above all a-political, while all forms of system-transcendent solutions are weeded out as being too political. Ironically, teaching about sustainable development thereby becomes politicised. There is therefore also a lack of critical perspectives on what knowledge about SD is in Swedish schools, and perspectives that do not support a dominant liberal discourse tend to be excluded.”
Reviewer 2 Report
I found this article very informative and important to read at this point in time. Given the variability of national curricula with regards to education for sustainability and the constraints that are placed on interdisciplinary learning, I found this article very insightful.
The study is well presented; perhaps more could have been said about the International literature on teachers' perception of education for sustainability so that the authors could discuss more in detail the activist dimension they mention in the conclusions.
overall, this is a very good piece of work.
Author Response
The study is well presented; perhaps more could have been said about the International literature on teachers' perception of education for sustainability so that the authors could discuss more in detail the activist dimension they mention in the conclusions.
We have updated the literature and we also have an extended discussion under “Conclusions":
- Meisert, A. & Böttcher, F. (2019). Towards a discourse-based understanding of sustainability education and decision making. Sustainability, 11(21), 5902.
- Sund, P. & Gericke, N. (2020). Teaching contributions from secondary school subject areas to education for sustainable development – a comparative study of science, social science and language teachers. Environmental Education Research, 26(6), 772-794.
- Misiaszek, G. W. (2019). Ecopedagogy: teaching critical literacies of ‘development’, ‘sustainability’, and ‘sustainable development’. Teaching in Higher Education, 25(5), 615-632.
“Conclusions
This study shows that the participating teachers lack a positive vision for a future sustainable society, at the same time as it seems to be taboo to talk about what an unsustainable society might actually mean in the long run. Furthermore, presentations of environmental problems - and knowledge about their causes - always need to be combined with instructions for solutions, which means that the teachers thereby impose on their pupils self-sacrifice for the sake of other people, future generations, other cultures or other species. Thus, more progressive knowledge and actions that may be highly important for contributing to a future sustainable society are not raised, for example, de-scaling, eco-towns and other alternative societal visions. System-immanent solutions are in this context understood as unproblematic, constructive and above all a-political, while all forms of system-transcendent solutions are weeded out as being too political. Ironically, teaching about sustainable development thereby becomes politicised. There is therefore also a lack of critical perspectives on what knowledge about SD is in Swedish schools, and perspectives that do not support a dominant liberal discourse tend to be excluded.”
Reviewer 3 Report
The paper provides an important and insightful study of problems related to teachers' handling of questions related to sustainability within school curricula. Although it is beyond the scope of this paper, I wonder if the authors have any plans for examining ways in which these problems might be addressed.
Author Response
Important comment! We have tried to develop this kind of discussion a bit in the conclusions:
“Conclusions
This study shows that the participating teachers lack a positive vision for a future sustainable society, at the same time as it seems to be taboo to talk about what an unsustainable society might actually mean in the long run. Furthermore, presentations of environmental problems - and knowledge about their causes - always need to be combined with instructions for solutions, which means that the teachers thereby impose on their pupils self-sacrifice for the sake of other people, future generations, other cultures or other species. Thus, more progressive knowledge and actions that may be highly important for contributing to a future sustainable society are not raised, for example, de-scaling, eco-towns and other alternative societal visions. System-immanent solutions are in this context understood as unproblematic, constructive and above all a-political, while all forms of system-transcendent solutions are weeded out as being too political. Ironically, teaching about sustainable development thereby becomes politicised. There is therefore also a lack of critical perspectives on what knowledge about SD is in Swedish schools, and perspectives that do not support a dominant liberal discourse tend to be excluded.”